# Bionomics and Ecological Services of Megaloptera Larvae (Dobsonflies, Fishflies, Alderflies)

**DOI:** 10.3390/insects10040086

**Published:** 2019-03-27

**Authors:** Sara Lariza Rivera-Gasperín, Adrian Ardila-Camacho, Atilano Contreras-Ramos

**Affiliations:** 1Instituto de Biología, UNAM, Depto. de Zoología, 04510 Ciudad de México, Mexico; zaralariza@gmail.com; 2Posgrado en Ciencias Biológicas, UNAM, Sede Instituto de Biología, 04510 Ciudad de México, Mexico; aardilac88@gmail.com

**Keywords:** immature insects, hellgrammites, insect diversity, fossil record, phylogeny, ecosystem services, adaptation, predation

## Abstract

Megaloptera belong to a large monophyletic group, the Neuropteroidea, together with Coleoptera, Strepsiptera, Raphidioptera, and Neuroptera. With the latter two, this order constitutes the Neuropterida, a smaller monophyletic subset among which it is the only entirely aquatic group, with larvae of all species requiring submersion in freshwater. Megaloptera is arguably the oldest extant clade of Holometabola with aquatic representatives, having originated during the Permian before the fragmentation of Pangea, since about 230 Ma. It includes 54 genera (35 extant and 19 extinct genera), with 397 extant described species and subspecies. Recent Megaloptera are divided into two families: Corydalidae (with subfamilies Corydalinae—dobsonflies and Chauliodinae—fishflies) and Sialidae (alderflies), both widely yet disjunctively distributed among zoogeographical realms. All species of Megaloptera have aquatic larvae, whereas eggs, pupae, and adults are terrestrial. The anatomy, physiology, and behavior of megalopteran larvae are specialized for an aquatic predatory habit, yet their ecological significance might still be underappreciated, as their role in food webs of benthic communities of many temperate and tropical streams and rivers is still understudied and largely unquantified. In many freshwater ecosystems, Megaloptera larvae are a conspicuous benthic component, important in energy flow, recycling of materials, and food web dynamics.

## 1. An Ancient Holometabolous Order

The order Megaloptera Latreille, 1802, has been traditionally regarded as an ancient holometabolous group [1,2], and arguably includes the oldest known representatives of aquatic insects with complete metamorphosis. They are conceded slight attention by general or applied entomologists, yet aquatic entomologists are aware that they can attain high abundances in particular ecosystems and regions. Megaloptera include some of the most impressive insects, because of their large size (e.g., adult male *Corydalus magnus* Contreras-Ramos, 1998 with ca. 180 mm wingspan [3], mature larvae of *Acanthacorydalis orientalis* (McLachlan, 1899) with up to 90 mm body length [4]), and the odd appearance of males with elongate mandibles in some New World *Corydalus* and Oriental *Acanthacorydalis* [5] or a head with postocular expansions in Central American *Platyneuromus* [6,7]. With a comparatively low diversity of less than 400 extant described species and subspecies, their ecological role might be underappreciated, as megalopteran larvae or hellgrammites (Figure 1) can be relevant members of food webs in benthic communities of many temperate and tropical lakes, streams, and rivers. The extant species of Megaloptera are divided into two families: Corydalidae Leach, 1815, and Sialidae Leach, 1815 (alderflies), both widely distributed in all zoogeographical realms. The family Corydalidae is further divided into two subfamilies: Corydalinae Davis, 1903 (dobsonflies), and Chauliodinae van der Weele, 1909 (fishflies). All species of Megaloptera have aquatic larvae, whereas eggs, pupae, and adults of all species are terrestrial (Figure 2). Megaloptera are considered one of the ancient groups within holometabolous insects [8,9].

## 2. Distribution and Diversity

The order Megaloptera contains 52 genera represented by 397 extant described species and subspecies, distributed in 35 genera, and 34 extinct described species distributed in 24 extant and extinct genera (Table 1 and Table 2) [8,10,11,12,13]. This order has a wide and fragmented geographical distribution, with Corydalinae (174 spp.) in the Americas, Asia, and South Africa; Chauliodinae (140 spp.) in North America, Argentina, Brazil, Chile, Australia, New Zealand, Madagascar, South Africa, Japan, and Southeast Asia; and Sialidae (83 spp.) in the Americas, Europe, Asia, South Africa, Madagascar, and Australia (Table 1) [5,14,15]. The main species richness centers of Corydalinae and Chauliodinae are China and Southeast Asia. A second region of high diversity in Corydalinae are Amazonia and the Andes Mountains. Chauliodinae is speciose in Australia and along the Pacific Coast of the United States, but is absent through most of the Neotropics, being restricted to Chile and southeastern Brazil [9]. Corydalidae are absent from Europe, the Middle East, Central Asia, tropical Africa, and boreal regions, with Corydalinae also absent from Australia [14]. Sialidae is most speciose in temperate regions and is absent from tropical Africa and portions of the Oriental region [14]. Several species are widespread (e.g., the South and Central American *Corydalus peruvianus* Davis, 1903, and the Amazonian *C. affinis* Burmeister, 1839), whereas others tend to be endemic and have restricted distributions (e.g., the western Mexican *Chloronia pallida* (Davis, 1903) or the Central American *Platyneuromus reflexus* Glorioso & Flint, 1984). Endemism in Megaloptera is fairly common, even at the genus level (e.g., *Chloroniella* in South Africa, *Dysmicohermes* and *Orohermes* in western North America, *Madachauliodes* in Madagascar, and *Apochauliodes* and *Austrosialis* in Australia). Regarding the taxonomic knowledge of the larval stages, mature larvae are described for six genera of Corydalinae (*Acanthacorydalis*, *Chloronia*, *Corydalus*, *Neoneuromus*, *Platyneuromus*, and *Protohermes*), 13 genera of Chauliodinae (*Apochauliodes*, *Archichauliodes*, *Chauliodes*, *Dysmicohermes*, *Madachauliodes*, *Neochauliodes*, *Neohermes*, *Nigronia*, *Orohermes*, *Parachauliodes*, *Platychauliodes*, *Protochauliodes*, and *Taeniochauliodes*), and seven genera (*Austrosialis*, *Haplosialis*, *Ilyobius*, *Indosialis*, *Leptosialis*, *Sialis*, and *Stenosialis*) of Sialidae [4,16,17,18].

## 3. Origin and Fossil Record

Insects underwent an explosive radiation during the Carboniferous and early Permian, especially with the origin of holometabolous insects (= Holometabola), which later became the dominant multicellular organisms in most terrestrial and freshwater ecosystems [2,19]. Recent studies suggest that Megaloptera species have existed since the Permian, before the fragmentation of Pangea, with subsequent radiation of Gondwanan and Laurasian elements, represented at present by numerous fossil taxa known from various Mesozoic deposits, many placed in now-extinct families (Table 2) [15,20].

The oldest fossils that have been described of Megaloptera correspond to representatives of the Russian-Mongolian family †Parasialidae, genera †*Parasialis* (279.3–272.3 Ma, Permian) and †*Sojanasialis* (272.3–268.8 Ma, Permian). The fossil record is older than the estimates of megalopteran origin of about 230 Ma proposed by Misof et al. [19], and more closely agrees with a recent estimation of a middle Permian origin [20]. The extinct Russian family †Nanosialidae was also described from the Permian (259.8–254.2 Ma), however, this family has been proposed as ancestral members of the snakefly lineage [21,22]. Larvae of Megaloptera are known in the middle Triassic of Grès des Vosges, France (André Nel, pers. comm.), whereas fossil genera of Sialidae, †*Dobbertinia* and †*Sharasialis*, are known from Old World Early and Late Jurassic deposits, respectively. The oldest representatives of Corydalidae belong to †*Jurochauliodes* and †*Eochauliodes* from the Mongolian-Chinese Middle Jurassic. However, splitting of the two main lineages, Corydalidae and Sialidae, might have been earlier, about 183 Ma during Middle Jurassic [10], or 210–215 Ma during late Triassic [19]. Other extinct corydalid genera are from Cretaceous deposits, both from the Old and New World (Table 2) [15,23]. More-recent species from extant and extinct genera have been described from the Cenozoic, mostly from the Eocene, Miocene, and Pliocene of Australia, Canada, Russia, Turkey, France, Germany, Dominican Republic, and the USA (Table 2) [23]. The neuropteroid family Corydasialidae was proposed from a specimen preserved in Baltic amber (late Eocene) [24]. A second species was assigned from a compressed forewing from the Ypresian (early Eocene) Okanagan Highlands, British Columbia, Canada [25]. This family is problematic because it possesses a generalized wing venation, and the characteristics used to place it within Megaloptera are not strongly diagnostic, yet a position in Megaloptera seems to be favored [25].

## 4. Phylogenetic Relationships

The order Megaloptera belongs to a large monophyletic group, the Neuropteroidea, together with the Coleoptera (beetles), Strepsiptera (twisted-wing parasites), Raphidioptera (snakeflies), and Neuroptera (lacewings). With the latter two, this order constitutes the Neuropterida, a smaller monophyletic subset, among which it is the only entirely aquatic group, with larvae of all species requiring submersion in freshwater [8,20,26]. For years, the sister group relationships within Neuropterida were contentious [2,27]; however, strong support has been provided recently that Megaloptera and Neuroptera are sister taxa [19,20,26].

It has long been assumed that the Megaloptera are a natural group (e.g., [27]), in part because of similarity in larval morphology and common ecology (e.g., a small pair of lateral filaments on the eighth abdominal segment, predatory habits, and aquatic habitat). However, its monophyletic origin was challenged by recent morphological and molecular studies (e.g., [10,28,29]). Nonetheless, recent genomic analyses have confirmed the monophyly of Megaloptera [11,13,19,20]. Despite an old hypothesis supporting a sister relationship with Chauliodinae + Sialidae [30,31], recent molecular analyses support the sister relationship of Corydalinae + Chauliodinae (i.e., a monophyletic Corydalidae sister to Sialidae) (e.g., [11]).

A phylogeny of Corydalinae based on morphological characters [32] (Figure 3a) recognized four main dobsonflies lineages: (1) the monotypic *Chloroniella* from South Africa (as sister to all other lineages), much in agreement with a former analysis [33], except the latter placed *Acanthacorydalis* as sister to the New World lineage and *Chloroniella* was not included; (2) the *Protohermes* lineage that includes the primarily Oriental genera *Protohermes* and *Neurhermes;* (3) the *Corydalus* lineage, which includes the New World genera *Corydalus*, *Platyneuromus*, and *Chloronia*; and (4) the *Nevromus* lineage composed of the Oriental genera *Nevromus*, *Neoneuromus*, and *Acanthacorydalis*.

A phylogenetic analysis of Chauliodinae that included all fossil and extant genera [9] (Figure 3b) recognized three major clades within fishflies: (1) the *Dysmicohermes* lineage, which includes the western North American genera *Dysmicohermes* and *Orohermes*; (2) the *Protochauliodes* lineage, which includes *Madachauliodes* from Madagascar, *Taeniochauliodes* from South Africa, *Neohermes* from North America, *Nothochauliodes* from Chile, and the disjunctly distributed genus *Protochauliodes* from western North America, Chile, and Australia, as well as extinct fossil taxa found in Asia; and (3) the diverse *Archichauliodes* lineage, which includes *Platychauliodes* from South Africa, *Chauliodes* and *Nigronia* from North America, *Apochauliodes* from western Australia, *Ctenochauliodes*, *Neochauliodes*, *Parachauliodes*, and *Sinochauliodes* from the Oriental region, and the disjunct genus *Archichauliodes* from South America and Australia. In addition, Cardoso-Costa et al. [34] described the new monotypic genus *Puri* from Brazil, which appears to be a member of the *Archichauliodes* lineage [35].

The first global phylogeny of Sialidae was based on morphological data [36] (Figure 3c). Its authors recognized two subfamilies: †Sharasialinae and Sialidinae. Within Sialidinae, the genus *Austrosialis* is sister to all other extant genera, an assemblage which includes three monophyletic lineages: (1) the *Stenosialis* lineage composed of *Stenosialis* from Australia and *Leptosialis* from South Africa; (2) the *Ilyobius* lineage composed of *Ilyobius* from the Neotropics, *Haplosialis* from Madagascar and *Indosialis* from Turkey, India, and Singapore; and (3) the *Sialis* lineage, which includes the species-rich genus *Sialis* from the Nearctic, Palearctic, and Oriental regions and the American *Protosialis*.

## 5. Evolution of Ecological Roles

As a member of the Neuropterida, Megaloptera maintain a generalized life history, with noteworthy differences in habitat and food preference only between families. In general, larval Megaloptera are voracious predators with some known to be scavengers; cannibalism might occur in all species [3,5]. Several alderfly species inhabit lentic habitats, or depositional zones of lotic systems, where they feed on small prey they can capture within the soft sediments (e.g., oligochaetes, chironomids, ostracods, or other small invertebrates). Dobsonfly larvae inhabit lotic waters, as most fishfly species do, except a few are associated with lentic environments (e.g., the North American *Chauliodes*). Yet, their polyphagous habits are a generalized trait for all Megaloptera, except perhaps for alderfly larvae associated with soft sediments (e.g., bottoms of lakes) and smaller prey, and one alderfly species has been reported as a collector-gatherer [37]. Larval *Sialis* consume small prey without chewing [5,38], but larger prey items are captured with mandibles and forelegs, a behavior that has also been observed in *Corydalus* [39]. It has been observed in captivity that once the prey is grasped, the larva manipulates it with the forelegs and the mouthparts to cut and swallow small portions [39]. Larval Corydalidae are ambush predators and are typically found under stones, submerged plant materials, roots, and logs. Their prey spectrum includes a wide variety of aquatic organisms, including other immature aquatic insects, crustaceans, worms, and tadpoles [5]. In laboratory conditions, mature larvae of *Corydalus* accepted different types of live and dead terrestrial arthropods, earthworms, small fishes, and even fresh fish meat as food. These predatory larvae remain motionless for long periods of time and often move to other suitable places during the night [40], probably searching for prey, as substrate type is correlated with the success of predation [41]. Size of the prey captured is positively correlated with the size of the mandibles, and consequently, with the size of the larva [42].

The ancestral environment for Megaloptera probably was lotic, with several alderfly species colonizing lentic habitats, particularly in temperate areas (e.g., Neotropical alderflies are mostly lotic); dobsonflies tend to inhabit tropical lotic habitats, whereas fishflies tend to be temperate [43], with at least the North American genus *Chauliodes* sometimes inhabiting lentic waters, including tree holes [44]. Several corydalid species inhabit intermittent streams, such as western North American fishflies *Neohermes californicus* (Walker, 1853), *N. filicornis* (Banks, 1903), *Protochauliodes aridus* Maddux, 1954, and *P. spenceri* Munroe, 1953 [43,45], or the dobsonflies *Corydalus luteus* Hagen, 1861, from Middle America and *C. affinis* Burmeister, 1839, from the Argentinean Chaco [3,46]. More specific relationships of habitat preference and geography appear to exist, with a gradient of species from those that are eurytopic and widespread to others that are stenotopic and endemic. Examples of the latter in Suriname include *Corydalus nubilus* Erichson, 1848, which appears to be confined to large open rivers, and *C. batesii* McLachlan, 1868, which seems to be restricted to creeks shaded by small bushes [47]. No analysis has been made yet globally to compare habitat restriction to pollution tolerance.

## 6. Essential Services for Ecosystem Function

Megaloptera larvae are found in clean lakes, ponds, and watercourses; however, some species tolerate some water pollution or anthropogenic eutrophication. In several freshwater ecosystems, these insects are a conspicuous component, important in energy flow, recycling of materials, and food chains [3,48]. The significance of the ecological role of larval Megaloptera rests upon their euryphagic, predatory habits, although many are probably also scavengers. These traits, particularly at higher population densities, make the group a significant element in trophic networks, and thus a contributor for the equilibrium of aquatic communities. Larvae of Corydalidae may be the largest and most abundant predator in fishless streams, and thus play an important role in food web dynamics [49]. Typically, the group regulates populations of other invertebrates, while also being an important prey for other invertebrates, fishes, and some terrestrial vertebrates [50]. Species of restricted distribution, often present in low population numbers, such as some Neotropical Sialidae, might play an important role for the conservation of their habitats [51].

### Food for Other Organisms and Uses for Humans

Larval Megaloptera represent an important component of aquatic food webs [52] and constitute part of the diet of an important variety of organisms, including vertebrates and other invertebrates. Odonata naiads and predatory Trichoptera have been reported among the aquatic invertebrate enemies of Megaloptera, yet many other groups such as aquatic Coleoptera and Hemiptera may also prey on megalopteran larvae. Predatory fishes are probably the principal natural enemies of this group [53]. In South America, some species of the rodent *Ichtyomys* prey on mature larvae of *Corydalus* and *Chloronia*, and in Mexico pigs forage on *Corydalus* larvae [5,54]. In Mexico and other regions of Latin America, hellgrammites, particularly of the genus *Corydalus,* are used as food by human communities, and in North America are appreciated as fishing bait [50]; *Acanthacorydalis* larvae are used in traditional food and medicine in southwestern China [4], and dehydrated *Protohermes* larvae are used as infant tranquilizer in Japan [33]. Energy flow from the aquatic towards the terrestrial environment, based on considerable figures of secondary production by megalopteran larvae [55,56], is another understudied service of this biological group in neighboring freshwater and terrestrial ecosystems. At least one artist (Adam Pasamanick) has built a realistic bronze sculpture of a male *Corydalus cornutus*, now exhibited at the Fairbanks Museum and Planetarium in Vermont (Figure 4).

## 7. Anatomical, Physiological, and Behavioral Adaptations for Life in Water

Within Neuropterida, Megaloptera are remarkable because of their long-lived larval stage, which is exclusively adapted to an aquatic life style. Only two of 15 families of Neuroptera, Nevrorthidae, and Sisyridae, have aquatic larvae, and all Raphidioptera have an entirely terrestrial life cycle [20]. The anatomy, physiology, and behavior of the megalopteran larva are specialized for an aquatic predatory habit. Each of the higher taxa (Sialidae, Chauliodinae, Corydalinae) occupy at most slightly differentiated spatial and functional niches. Larval sialids are small (10–20 mm body length), typically live for 1–2 years, and are found in both lotic and lentic habitats. Larval corydalids may be large (20–80 mm body length), and long-lived (2–5 years). Corydalids are commonly found in mountain streams, but have been found in a wide variety of aquatic habitats including large rivers, swamps, and water-filled vegetation [35].

Megaloptera larvae have variously patterned or somewhat uniformly colored heads and thoracic nota (mostly pronota). A pattern, because of muscle attachment marks, might serve as camouflage; however, larval coloration can be generally considered cryptic. In the New World, *Corydalus* dobsonfly larvae tend to be dusky-brown to blackish and live in dark, rocky substrates. Larvae of *Chloronia* species have a uniformly reddish color, sometimes with four dark dots on the pronotum, and they live in gravel, rock, or deposited organic matter of similar or darker tones. Larvae of *Platyneuromus* species are yellowish-patterned and live in pale to grayish sandy or rocky limestone and travertine substrates. Often the hellgrammite cuticle becomes impregnated with materials carried by their native streams (A.A.-C., A.C.-R., personal observations). The cephalic capsule is flattened, robust, and heavily sclerotized, and prognathous to subprognathous with well-developed chewing mouthparts. The mandibles are robust and sharply pointed, nearly as long as the larval head and equipped with two to four preapical sharp teeth, which are adapted for their predatory habits. The antennae are short, with four articles in Sialidae and Chauliodinae and five in Corydalinae; a peg organ with minute spine-like sensilla is present on the apical part of the second antennomere of Chauliodinae and a spine-like or conical sensilla is on the same region in Corydalinae and Sialidae. The musculature of the head has been described for *Corydalus cornutus* (L., 1758) [57] and for *Neohermes* sp. [58].

Larvae of Megaloptera are visually capable, as they have been observed to react to the sight presence of prey. Spectral sensitivity has not been investigated in the group, but responses to UV, blue, and green have been observed in larvae of *Corydalus* (A.A.-C., personal observation). The first-instar larva of *Sialis* is planktonic and positively phototactic; conversely, later instars are bottom-dwellers and negatively phototactic. The light sensitive organs are situated anterolaterally, just before the antennal insertions; they are composed of six fully developed stemmata, arranged either with five peripheral ones arranged as a pentagon around a central stemma (Corydalinae and Chauliodinae) or as curved, anterior and posterior rows (Sialidae and Chauliodinae). The retina of each stemma is concave and two-tiered with around 40 retinular cells (Sialidae) or composed of hundreds of retinular cells (Corydalinae) whose rhabdomeres are arranged as an irregular net [59,60]. Although these stemmata are probably not image-resolving organs, laboratory studies provide evidence for highly sensitive eyes (A.A.-C., personal observation).

The thorax is flattened and robust, with the prothorax longer than meso- and metathoraces and a heavily sclerotized and quadrangular pronotum; the basisternum is heavily sclerotized; the forelegs are slightly shorter than the mid- and hindlegs and are used for digging behavior for species that burrow into dry streambeds; each tarsus consists of a cylindrical basitarsus, which bears a pair of simple claws distally [5,57]. These traits may relate to a life in rocky streams with riffles and strong currents, the habitat of corydaline larvae.

The abdomen is elongate, 10-segmented, soft and muscular, flattened, and becoming narrower posteriorly. Paired lateral filaments (tracheal gills) are present on segments 1–7 (Sialidae) or 1–8 (Corydalidae), generally covered with rows of long sensilla, which may be related to sensing both prey and predators. Lateral filaments also have thin cuticle and a tracheal trunk with numerous branches, thus being involved in respiration, particularly in Sialidae and Chauliodinae [5]. Larvae also have two pairs of thoracic spiracles and eight pairs of abdominal spiracles for oxygen uptake directly from atmospheric air. Segment 10 of Sialidae has a long and tapered median filament, which is covered with long hair-like sensilla, whereas in Corydalidae there is a pair of prolegs, each of which bears a lateral filament and two hook-like claws to firmly attach to the substrate. Megaloptera larvae can swim; members of both families can make undulating movements with the abdomen, probably as a defensive response against predation and cannibalism, but also useful for dispersion [61]. Abdominal segment 8 of Chauliodinae has the spiracles located dorsally, sometimes at the ends of long respiratory tubes, which may be used as ‘snorkels’ to breathe at the water surface [62]. Some species of this subfamily are adapted to live in stagnant water, whereas others are found in shallow waters of lotic environments where they can either expose the respiratory tubes to atmospheric oxygen or stay under water, depending on oxygen concentrations [5]. Generally, larvae of Corydalinae are adapted to well-oxygenated streams; they have tripartite ventral tufts of tracheal gills close to the ventral base of each lateral filament on the first seven abdominal segments; these gills are involved in gas exchange, and it is believed they do not participate in osmoregulation, but this role is taken by the hindgut [5]. The abdominal cuticle has a layer of highly modified setae of several types, which may be classified as macro- or microsetae. Corydalinae larvae have both types, with macrosetae erect and varying in morphology, often subglobose and canaliculate with open tips; microsetae are generally star- or scale-shaped and decumbent. These modified setae are probably involved in chemoreception or sensing the current flow, but physiological research on these structures is lacking. Most fishfly and alderfly larvae lack thick tergal macrosetae, but some Chauliodinae genera have thin, subglobose, and apparently closed macrosetae [35,63].

## 8. Responses to Threats of Water Pollution and Climate Change

Some of the strongest threats to megalopteran larvae are pollution, eutrophication, and other forms of freshwater and riparian habitat degradation, as a consequence of unregulated human population growth, including expansion of urban areas, deforestation, and water pollution from different sources, such as agricultural production (fertilizers and pesticides from crop production, sewage from animal operations), sewage from pets, petrochemical runoff from motorways, sediment from construction activities, sediment from bank erosion in storm surges caused by rapid runoff from impervious surfaces, and more [64]. Several species with widespread distribution appear to tolerate moderate habitat disturbance, such as *Corydalus cornutus* in North America, *C. luteus* in Middle America, and *C. armatus* in South America [46,65] (A.A.-C., A.C.-R., personal observations), yet formal documentation of the degree of resistance is still incomplete. Other species are more restricted in distribution and limit their presence to pristine conditions, and probably are intolerant and vulnerable to habitat alteration. Invasive species like rainbow trout (*Oncorhynchus mykiss* (Walbaum, 1972)) and red swamp crayfish (*Procambarus clarkii* (Girard, 1852)) may be significant threats to Megaloptera populations, either by competing with or preying on hellgrammites. These species have become the subject of recent research in the Tropical Andes due to their negative effect on fresh water systems, as they are highly aggressive and voracious predators, and spread pathogens to the native fauna [66,67]. Megaloptera may also be useful for environmental monitoring and distribution modeling to track and predict their future distributions under various climate change scenarios. Being conspicuous insects, both during their aquatic larval and terrestrial egg/pupal/adult stages, they might also be appropriate subjects for insect conservation studies, and thus could potentially be incorporated into management initiatives for habitat and biodiversity protection.

## 9. Conclusions

Despite its low global diversity of close to 400 described species, Megaloptera is an important and conspicuous component of many benthic macroinvertebrate communities, either in tropical or temperate latitudes, as well as in lentic and lotic systems. Often thought of as representatives of an ancient holometabolous lineage, with generalized predatory habits, detailed studies of behavior and sensory capacities of megalopteran larvae are quite scarce. Estimations of their productivity and population density, particularly in tropical environments, are also neglected areas of research. Modern approaches to larval behavior may prove that these insects have more sophisticated capabilities to locate and capture prey than previously realized. Recently, a fairly complete scheme of alpha taxonomy and phylogeny has emerged for this group, although specific phylogenetic issues have yet to be tested or corroborated with cutting-edge molecular techniques; many species level larval descriptions are still missing. Tolerance values of larvae of many species are unknown, and there is no comparative global database from reliable empirical sources. The cultural value and relevance of the group as a source of human food offer opportunities for original research in ethnozoology and novel findings of potential resources.

## Figures and Tables

**Figure 1 insects-10-00086-f001:**
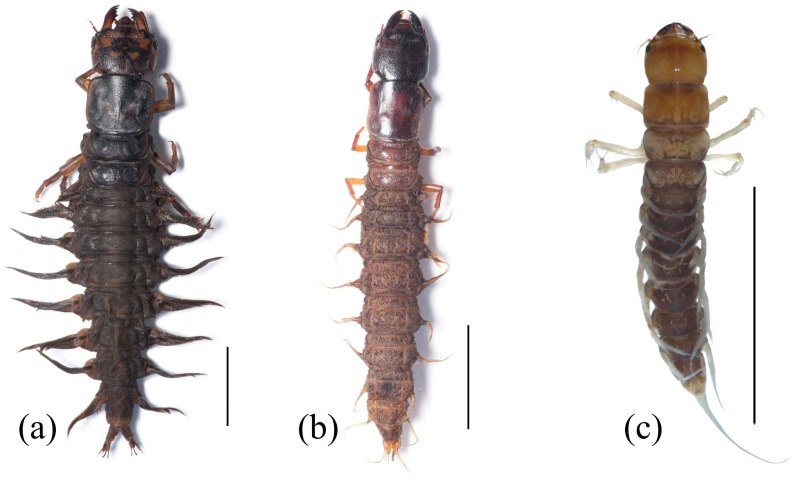
Habitus of Megaloptera larvae. (**a**) *Corydalus liui* Ardila-Camacho & Contreras-Ramos, 2018, (**b**) *Chloronia* sp., (**c**) *Ilyobius* sp. Scale = 10 mm. Images by Adrian Ardila-Camacho.

**Figure 2 insects-10-00086-f002:**
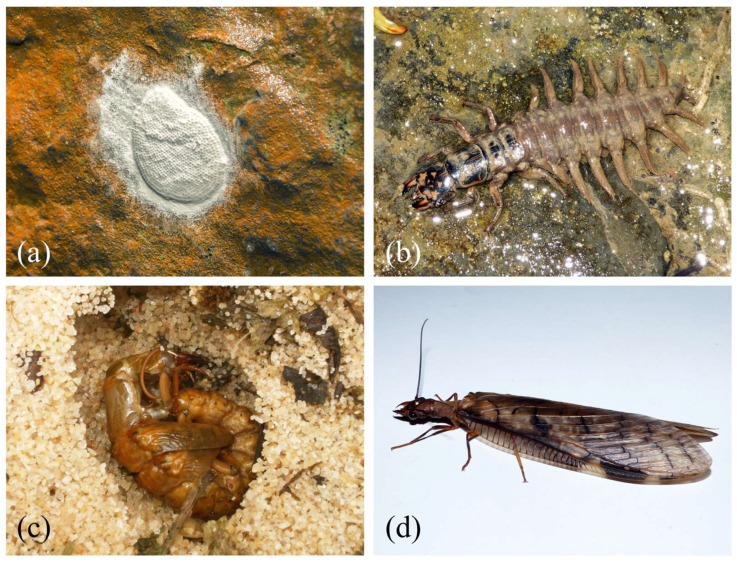
Developmental stages of *Corydalus*. (**a**) egg mass of *Corydalus* sp., (**b**) mature larva of *Corydalus liui*, (**c**) pupa of *Corydalus peruvianus* Davis, 1903, (**d**) female of *Corydalus liui*. Images by Adrian Ardila-Camacho.

**Figure 3 insects-10-00086-f003:**
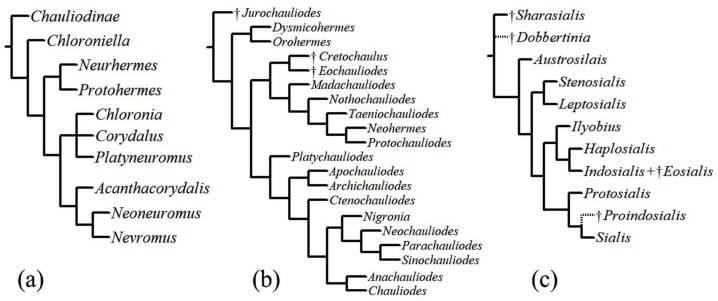
Phylogenetic trees. (**a**) phylogeny of Corydalinae proposed by Contreras-Ramos [32], (**b**) phylogeny of Chauliodinae proposed by Liu et al. [9], (**c**) phylogeny of Sialidae by Liu et al. [36] (putative positions indicated by dotted lines).

**Figure 4 insects-10-00086-f004:**
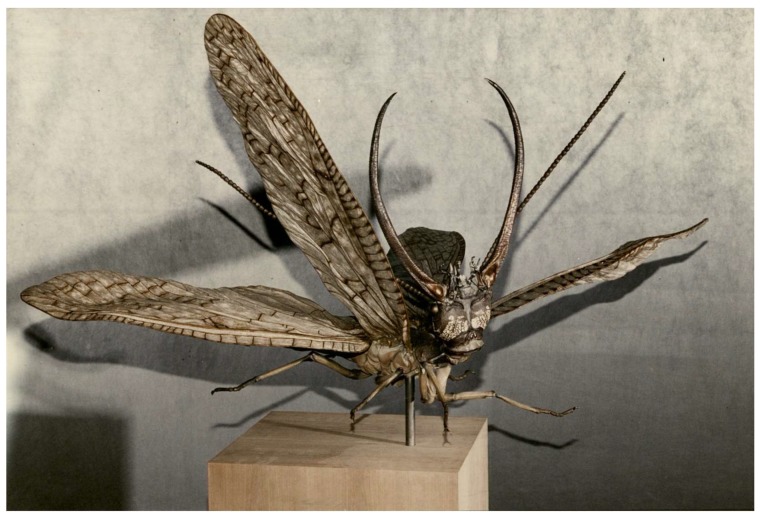
Bronze sculpture of adult dobsonfly, *C. cornutus* (L.), exhibited at the Fairbanks Museum and Planetarium in Vermont, United States (artist Adam Pasamanick, dimensions 1.82 m high × 1.52 m wide × 1.21 m long).

**Table 1 insects-10-00086-t001:** Genera of Megaloptera with extant valid species (* = includes subspecies, ** = possible erroneous combination of an Oriental species).

Genus	Number of Species	Biogeographic Region
CORYDALIDAE: Corydalinae		
*Acanthacorydalis* van der Weele, 1907	8	Oriental
*Chloronia* Banks, 1908	18	Nearctic, Neotropical
*Chloroniella* Esben-Petersen, 1924	1	Afrotropical
*Corydalus* Latreille, 1802	39	Nearctic, Neotropical, Oriental **
*Neoneuromus* van der Weele, 1909	14	Oriental, Palaearctic
*Neurhermes* Navás, 1915	7	Oriental
*Nevromus* Rambur, 1842	6	Oriental
*Platyneuromus* van der Weele, 1909	3	Nearctic, Neotropical
*Protohermes* van der Weele, 1907	78	Oriental, Palaearctic
CORYDALIDAE: Chauliodinae		
*Anachauliodes* Kimmins, 1954	1	Oriental
*Apochauliodes* Theischinger, 1983	1	Australian
*Archichauliodes* van der Weele, 1909	21	Australian, Neotropical
*Chauliodes* Latreille, 1802	2	Nearctic
*Ctenochauliodes* van der Weele, 1909	13	Oriental
*Dysmicohermes* Munroe, 1953	2	Nearctic
*Madachauliodes* Paulian, 1951	3	Afrotropical
*Neochauliodes* van der Weele, 1909	47	Oriental, Palaearctic
*Neohermes* Banks, 1908	6	Nearctic
*Nigronia* Banks, 1908	2	Nearctic
*Nothochauliodes* Flint, 1983	1	Neotropical
*Orohermes* Evans, 1984	1	Nearctic
*Parachauliodes* van der Weele, 1909	7	Oriental, Palaearctic
*Platychauliodes* Esben-Petersen, 1924	3	Afrotropical
*Protochauliodes* van der Weele, 1909	17 *	Australian, Nearctic, Neotropical
*Puri* Cardoso-Costa et al., 2013	1	Neotropical
*Sinochauliodes* Liu & Yang, 2006	4	Oriental
*Taeniochauliodes* Esben-Petersen, 1924	8	Afrotropical
SIALIDAE: Sialidinae		
*Austrosialis* Tillyard, 1919	2	Australian
*Haplosialis* Navás, 1927	2	Afrotropical
*Ilyobius* Enderlein, 1910	9	Neotropical, Palaearctic
*Indosialis* Lestage, 1927	3	Oriental, Palaearctic
*Leptosialis* Esben-Petersen, 1920	2	Afrotropical
*Protosialis* van der Weele, 1909	3	Nearctic, Neotropical
*Sialis* Latreille, 1802	60	Nearctic, Oriental, Palaearctic
*Stenosialis* Tillyard, 1919	2	Australian
Total: 35 genera	397 spp.	

**Table 2 insects-10-00086-t002:** Genera of Megaloptera with extinct valid species (BRA = Brazil, CAN = Canada, CHN = China, DEU = Germany, DOM = Dominican Republic, FRA = France, MNG = Mongolia, RUS = Russia, TUR = Turkey, USA = United States; * = some species from amber deposits; †: extinct taxon).

Genus	Species	Locality-Biogeographic Region	Period (Ma)
CORYDALIDAE: Corydalinae
† *Corydalites* Scudder, 1878	1	USA-Nearctic	Cenozoic (66 Ma)
*Corydalus* Latreille, 1802	1	DEU-Palaearctic	Late Jurassic (152 Ma)
† *Cratocorydalopsis* Jepson & Heads, 2016	1	BRA-Neotropical	Early Cretaceous (115 Ma)
† *Lithocorydalus* Jepson & Heads, 2016	1	BRA-Neotropical	Early Cretaceous (115 Ma)
CORYDALIDAE: Chauliodinae
*Chauliodes* Latreille, 1802	2 *	RUS-Palaearctic	Middle Eocene (47.8 Ma)
† *Chauliosialis* Ponomarenko, 1976	1	RUS-Palaearctic	Late Cretaceous (89.8 Ma)
† *Cretochaulus* Ponomarenko, 1976	1	RUS-Palaearctic	Early Cretaceous (139.4 Ma)
† *Eochauliodes* Liu et al., 2012	1	CHN-Oriental	Middle Jurassic (168.3 Ma)
† *Jurochauliodes* Wang & Zhang, 2010	1	CHN-Oriental	Middle Jurassic (168.3 Ma)
† CORYDASIALIDAE
† *Corydasialis* Wichard et al., 2005	1 *	RUS-Palearctic	Late Eocene (33.9–37.8 Ma)
† *Ypresioneura* Archibald & Makark., 2015	1	CAN-Nearctic	Early Eocene (47.8–56 Ma)
† NANOSIALIDAE: † Nanosialinae
† *Hymega* Shcherbakov, 2013	1	RUS-Palaearctic	Late Permian (259.8 Ma)
† *Lydasialis* Shcherbakov, 2013	1	RUS-Palaearctic	Late Permian (259.8 Ma)
† *Nanosialis* Shcherbakov, 2013	2	RUS-Palaearctic	Late Permian (259.8 Ma)
† *Raphisialis* Shcherbakov, 2013	1	RUS-Palaearctic	Late Permian (259.8 Ma
† PARASIALIDAE
† *Parasialis* Ponomarenko, 1977	4	RUS, MNG-Palaearctic	Permian (279.3 Ma)
† *Sojanasialis* Ponomarenko, 1977	1	RUS-Palaearctic	Middle Permian (272.3 Ma)
SIALIDAE: † Sharasialinae
† *Sharasialis* Ponomarenko, 2012	1	MNG-Palaearctic	Late Jurassic (163.5 Ma)
SIALIDAE: Sialidinae
† *Dobbertinia* Handlirsch, 1920	1	DEU-Palaearctic	Early Jurassic (182.7 Ma)
† *Eosialis* Nel et al., 2002	1	FRA-Palaearctic	Early Eocene (56 Ma)
*Ilyobius* Enderlein, 1910	3 *	DOM-Neotropical, RUS-Palearctic	Eocene, Miocene (47.8–5.3 Ma)
*Indosialis* Lestage, 1927	1	TUR-Palaearctic	Oligocene (33.9 Ma)
† *Proindosialis* Nel, 1988	1	FRA-Palaearctic	Late Miocene (7.2 Ma)
*Sialis* Latreille, 1802	4 *	FRA, DEU-Palaearctic	Eocene, Miocene, Pliocene (47.8–2.5 Ma)
Total: 24 genera	34 spp.

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
