# Peer review of "Bionomics and Ecological Services of Megaloptera Larvae (Dobsonflies, Fishflies, Alderflies)"

_insects, 2019, doi:10.3390/insects10040086_

Round 1
Reviewer 1 Report
The authors did an excellent job summarizing the biology of the Megaloptera. I especially appreciated the new insights on anatomical and physiological adaptations to aquatic habitats, including the portion on visual perception. I have just a few recommendations for improvement to the coverage and organization; most of my recommendations are related to minor grammatical suggestions to improve clarity. My recommendations are included as comments directly on the pdf file.

Author Response
We went over the pdf file and checked all indications and suggestions, accepting nearly all. An exception was that we preferred “ambushing predators” in reference to larval corydalidae (line 152), instead of “ambush predators”; also we did not add Neohermes inexpectatusas an example of inhabitant of intermittent streams (line 166), as this is a supposition (most likely well founded) by the reviewer and we could not support it by a reference. We feel we essentially complied with all indications by reviewer 1 and we thank him/her for all suggestions and encouragement.
Reviewer 2 Report
a nice paper well documented, only a few remarks concerning the fossils and spelling (some concepts to avoid as ancestor, primitive, basal, living fossils)
Megaloptera is the oldest clade within aquatic representatives of Holometabola
Be prudent, il is not because the recent taxa in this clade are aquatic that the representatives of the stem group are also aquatic
Line 15: reference ?, there are papers on the phylogeny Misof et al 2016, with dating, maybe indicate them
And the oldest known fossils are not the oldest representatives of the stem group ….
Line 15; genera extant and/or fossils, precise
Line 30: ancient is not synonym of ‘primitive’, and avoid primitive, it has no sense at all for taxa
Line 31: they certainly are the oldest KNOWN representativeS among aquatic insects
Line 46: Current species of Megaloptera are considered as one of the basal groups within holometabolous
No, they are not basal at all, Hymenoptera are the sister group of other Holometabola, and Megaloptera nest in the Neuropterida
living fossil is a non-sense, Megaloptera have evolved since the Permian and Triassic …. Remove
line 77: putative stem group Megaloptera, in fact these Permian taxa need revision using a phylogenetic tool
line 86: larvae of Megaloptera are known in the Mid Triassic of Grès des Vosges, France (Nel andré, pers. comm.)
line 161: ancestral of the crown group Mealoptera, for the stem group, we cannot know as we do not have larvae of the palaeozoic taxa.
Shcherbakov, D.E. 2013. Permian ancestors of Hymenoptera and Raphidioptera. ZooKeys, 358: 45-67.
This author that you cite for the taxa, supposed that the Permian families could be ‘ancestors’ of hymenoptera and/or Raphidioptera, you should discuss a little this point, at least to say that the positions of the Permian ‘megaloptera’ remain uncertain
Author Response
A nice paper well documented, only a few remarks concerning the fossils and spelling (some concepts to avoid as ancestor, primitive, basal, living fossils).
We agree, we avoided or moderated the use of these and similar terms, which indeed are subjective or difficult to define.
Megaloptera is the oldest clade within aquatic representatives of Holometabola. Be prudent, it is not because the recent taxa in this clade are aquatic that the representatives of the stem group are also aquatic.
Indeed, we agree, we neontologists seldom incorporate fossil “thinking” into our statements; many studies if not most phylogenetic research are based on extant taxa (e.g., molecular phylogenies) and make inference about time of lineage splitting, origin of taxa, etc., so we are quite used to this type of statements, we appreciate the warning. In our defense we may say that if all member of a an extant group share a trait, particularly when it could be considered a synapomorphy, we may assume the ancestor had the trait. In any event, we added the term “arguably” to temper the statement (as suggested by one of the editors).
Line 15: reference ?, there are papers on the phylogeny Misof et al 2016, with dating, maybe indicate them. And the oldest known fossils are not the oldest representatives of the stem group ….
Line 15; genera extant and/or fossils, precise…
The paper by Misof et al. (2014), a highly relevant insect phylogenomics reference, is indeed included, we also added “since about 230 Ma”. We corrected the indication for line 15, being explicit on numbers for extant and extinct taxa.
Line 30: ancient is not synonym of ‘primitive’, and avoid primitive, it has no sense at all for taxa.
We agree, the term primitive is more appropriate for specific characters in the context of a phylogeny. The term was deleted.
Line 31: they certainly are the oldest KNOWN representatives among aquatic insects.
The statement was modified, with …“and arguably includes the oldest known representatives of aquatic insects with complete metamorphosis”.
Line 46: Current species of Megaloptera are considered as one of the basal groups within holometabolous. No, they are not basal at all, Hymenoptera are the sister group of other Holometabola, and Megaloptera nest in the Neuropterida. Living fossil is a non-sense, Megaloptera have evolved since the Permian and Triassic …. Remove.
Indeed, the term basal is often misused; it actually refers to ancestral or ancestor, which is a quite difficult hypothesis to prove; we agree a more correct usage is as sister group. We have removed the term, instead we used …“are considered as one of the ancient groups within holometabolous insects” (line 49 revised version). We removed the term living fossil.
line 77: putative stem group Megaloptera, in fact these Permian taxa need revision using a phylogenetic tool.
We moderated the statement: …“Recent studies suggest thatMegaloptera species have existed since the Permian…”.
line 86: larvae of Megaloptera are known in the Mid Triassic of Grès des Vosges, France (Nel andré, pers. comm.)
We incorporated this information with the same reference (lines 102-103 revised version).
line 161: ancestral of the crown group Mealoptera, for the stem group, we cannot know as we do not have larvae of the palaeozoic taxa.
We agree, yet we often have to make inference or build hypotheses based on extant information, but yes, explicit evidence would prove such an ancestral habitat, or at least support such as hypothesis; we have stated this as a supposition: …“The ancestral environment for Megaloptera probably was lotic”… (line 191 revised version).
Shcherbakov, D.E. 2013. Permian ancestors of Hymenoptera and Raphidioptera. ZooKeys, 358: 45-67.
This author that you cite for the taxa, supposed that the Permian families could be ‘ancestors’ of hymenoptera and/or Raphidioptera, you should discuss a little this point, at least to say that the positions of the Permian ‘megaloptera’ remain uncertain
We went over this section on fossil taxa, added this reference (Shcherbakov 2013), and made a brief statement about Nanosialidae (lines 96-102 revised version).